# Attributable Mortality of Hip Fracture in Older Patients: A Retrospective Observational Study

**DOI:** 10.3390/jcm9082370

**Published:** 2020-07-24

**Authors:** Lorène Zerah, David Hajage, Mathieu Raux, Judith Cohen-Bittan, Anthony Mézière, Frédéric Khiami, Yannick Le Manach, Bruno Riou, Jacques Boddaert

**Affiliations:** 1Department of Geriatric Medicine, Hôpital la Pitié-Salpêtrière, Assistance Publique-Hôpitaux de Paris (APHP), 75013 Paris, France; judith.cohen-bittan@aphp.fr (J.C.-B.); jacques.boddaert@aphp.fr (J.B.); 2UMRS INSERM 1136, Institut Pierre Louis d’Epidémiologie et de Santé Publique, Sorbonne Université, 75013 Paris, France; david.hajage@aphp.fr; 3Department of Biostatistics, Hôpital la Pitié-Salpêtrière, APHP, 75013 Paris, France; 4UMRS INSERM 1158, Sorbonne Université, 75013 Paris, France; mathieu.raux@aphp.fr; 5Department of Anesthesiology and Critical Care, Hôpital la Pitié-Salpêtrière, APHP, 75013 Paris, France; 6Department of Rehabilitation, Hôpital Charles Foix, APHP, 94200 Ivry sur Seine, France; anthony.meziere@aphp.fr; 7Department of Orthopedic Surgery and Trauma, Hôpital La Pitié-Salpêtrière, APHP, 75013 Paris, France; Frederic.khiami@aphp.fr; 8Departments of Anesthesia & Health Research Methods, Evidence & Impact (HEI), Michael DeGroote School of Medicine, Faculty of Health Sciences, McMaster University, Hamilton, ON L8S 4L8, Canada; yannick.lemanach@phri.ca; 9UMRS INSERM 1166, IHU ICAN, Sorbonne Université, 75013 Paris, France; bruno.riou@aphp.fr; 10Department of Emergency Medicine and Surgery, Hôpital La Pitié-Salpêtrière, APHP, 75013 Paris, France; 11UMR INSERM U1135, Sorbonne Université, 75013 Paris, France

**Keywords:** hip fracture, attributable risk, mortality, prognosis, elderly

## Abstract

Hip fracture (HF) in older patients is associated with a high six-month mortality rate. Several clinical conditions may affect outcome, including baseline characteristics, co-existing acute illnesses, perioperative factors, and postoperative complications. Our primary objective was to estimate the respective effect of these four domains on six-month mortality after HF. A retrospective observational study using a monocentric cohort of older patients was conducted. All patients ≥ 70 years old admitted to the emergency department for HF and hospitalized in our perioperative geriatric care unit from June 2009 to September 2018 were included. Among 1015 included patients, five (0.5%) were lost to follow-up, and 1010 were retained in the final analysis (mean age 86 ± 6 years). The six-month mortality rate was 14.8%. The six-month attributable mortality estimates were as follows: baseline characteristics (including age, gender, comorbidities, autonomy, type of fracture): 62.4%; co-existing acute illnesses (including acute events present before surgery that could result from the fracture or cause it): 0% (not significantly associated with six-month mortality); perioperative factors (including blood transfusion and delayed surgery): 12.3%; severe postoperative complications: 11.9%. Baseline characteristics explained less than two-thirds of the six-month mortality after HF. Optimizing patients care by improving management of perioperative factors and thus decreasing postoperative complications, could reduce by a maximum of one quarter of the six-month mortality rate after HF.

## 1. Introduction

Hip fracture (HF) is a major public health concern that has increasing socio-economic consequences [1]. Although its incidence is decreasing, the total number of HF cases is increasing in line with the aging population (more than 1.6 million people undergo hip fracture in the world every year) [2,3]. This frequent emergency condition is associated with a poor prognosis. Mortality is much higher in people after HF than in the general population with comparable age, and remains high for many months after HF [4]. Mortality rate at six months varies from 8 to 26% [5].

As compared with elective total hip replacements, for HF patients, the mortality is increased 6- to 15-fold [6]. This difference was thought to be explained by the high prevalence of pre-existing medical conditions in this population, which is known to be associated with poor medical outcomes [2,7,8,9]. However, even when patients are matched for age, sex, and preoperative comorbidities, the in-hospital mortality after HF remains six times higher than that observed after elective total hip replacement [10]. Moreover, we previously found that geriatric scores, including age, sex, and comorbidities gave poor results to predict short- (30 days) and long-term (six months) postoperative mortality [11]. We can assume that factors other than baseline characteristics have a great impact on survival, including, for example, delayed surgery, anesthesia type, transfusions, or postoperative delirium [7,8,9,12]. The demonstration that a dedicated clinical action plan can significantly improve the six-month mortality of elderly patients with HF, compared to patients admitted to the orthopedic surgery department, is in favor of this hypothesis [9]. However, we searched on PubMed, on 1st March, 2018 and on 1st December, 2019, if a study had attempted to quantify the attributable mortality of HF in the elderly, without success (((attributable mortality[Title/Abstract]) OR (averaged attributable fractions[Title/Abstract])) AND (hip fracture[Title/Abstract])). 

We hypothesized that understanding the respective influence on factors associated to six-month mortality after HF and identifying the modifiable factors with the highest impact would be an essential step that could indicate the directions for care improvement. We proposed to classify all these factors into four domains: baseline characteristics, co-existing acute illnesses, perioperative factors, and postoperative complications. The relation between these factors and postoperative mortality is complex, and a fuller understanding of the contribution of each factor is needed to develop a better predictive model for HF outcomes in older people.

Our primary objective was to evaluate the respective influence of these four domains on six-month mortality in patients after HF.

## 2. Materials and Methods

The database was declared to the French National Commission on Computing and Liberty (CNIL) of Assistance Publique-Hôpitaux de Paris (APHP) (no 20190426181554). This report follows the STROBE and TRIPOD recommendations (Appendix A) [13,14].

### 2.1. Study Design, Setting, and Participants

This retrospective observational study was conducted in the perioperative geriatric unit (UPOG) of an academic hospital. UPOG is part of a dedicated orthogeriatric care pathway defined as follows: (1) early alert from the emergency department (ED); (2) consideration of HF as requiring surgery as soon as feasible (i.e., 24 h a day); (3) rapid transfer to the UPOG after surgery; and (4) rapid transfer of stable patients to a dedicated rehabilitation unit [8]. The management strategy in UPOG was previously described [9] and is detailed in Appendix A. 

From June 2009 to September 2018 (111 months), all consecutive patients with HF admitted to the UPOG were evaluated for eligibility. Patients were included if their primary presentation was due to HF (first hospitalization in UPOG) and if they were ≥70 years old. Patients with metastatic or periprosthetic fractures were excluded. Patients were followed until death or six months after admission. Surviving patients or their family were seen in routine consultation at six months or were contacted and interviewed by telephone. Missing patients were tracked through health care providers, particularly general practitioners, or any identified acquaintances. 

### 2.2. Description of the 4 Domains

We classified factors that could be associated with HF mortality into 4 domains: (1) baseline characteristics (including comorbidities but also age, gender, frailty, chronic medications, autonomy, functional status, and type of fracture); (2) co-existing acute illnesses (including acute events present before surgery that could result from the fracture or cause it); (3) perioperative factors (including delay to surgery, type of anesthesia, hemodynamic stability, and bleeding); and (4) postoperative complications (Figure 1).

### 2.3. Data Collection

Since the opening of UPOG in 2009, we have created a dedicated research database that was prospectively supplemented by 3 senior geriatricians (J.B., J.C.-B., L.Z.), experts in orthogeriatrics, and that integrates all the data from the orthogeriatric care pathway for each patient.

The following variables, collected prospectively, by interviewing patients, their family members or their physicians and pharmacists during the hospital stay, defined baseline characteristics before HF: age, gender, home or nursing home living conditions, walking ability, previous medical history, chronic medications, and type of fracture (radiological definition by an orthopedic surgeon). Co-morbidity severity, frailty, and functional status were calculated prospectively by one of the 3 senior geriatricians during the hospital stay. Co-morbidity severity was assessed with the Cumulative Illness Rating Scale (CIRS) [15], because all comorbidity scores are equivalent in predicting mortality in this population [11]. Frailty was assessed with the Rockwood score [16]. Functional status was evaluated with the Activities of Daily Living (ADL) scale [17] and the Instrumental Activities of Daily Living (IADL) scales [18].

During the perioperative period, we prospectively recorded the surgical treatment, delay and duration of surgery, type of anesthesia provided (general vs. spinal), and the amount of blood transfusion administered. After surgery, delays to first sitting and first walking, the destination at discharge of UPOG (home or rehabilitation), and length of stay in acute care and rehabilitation departments were recorded.

All postoperative complications during the acute care period (not in rehabilitation) were prospectively recorded.

Co-existing acute illnesses and the severity of postoperative complications were the only 2 variables to have been classified retrospectively, before any statistical analysis. They were both adjudicated by 2 senior geriatricians (J.B. and L.Z.), independently reviewing the medical charts (Kappa score 0.90 for acute co-existing illnesses and 0.97 for Dindo-Clavien score). In case of disagreement, reconciliation was reached with a third independent senior expert (B.R.).

Co-existing acute illnesses were defined as acute events that could have promoted the fall (e.g., acute coronary syndrome, stroke) or acute events that could result from the fall (e.g., other traumatic lesions, rhabdomyolysis) requiring specific treatment and/or affecting the prognosis. By definition, these conditions had to be present before surgery, but no predefined list was established.

The severity of postoperative complications was adjudicated with the Dindo-Clavien classification [19]. Dindo-Clavien classes from 3 to 5 were considered as severe complications. With several postoperative complications, the highest Dindo-Clavien score defined the final postoperative score.

### 2.4. Statistical Analyses

The statistician who performed the analyses was an independent statistician who was not involved in data collection or the initial definition of study objectives. The statistical plan of the study was established by him, after discussion with the authors, before transmission of the data and the beginning of the analyses (Appendix A). As the database was prospectively supplemented, all the authors were “blinded” to the research question at the time of data collection. The study is based on all available patients during the study period and thus no a priori power calculation was conducted.

Data are presented as mean ± SD, median (25–75 interquartile range) for non-Gaussian distributed variables, or numbers (percentages). Comparison of quantitative variables between survivors and deceased patients involved unpaired Student t tests, or Mann-Whitney tests in case of rejection of the normality assumption in one or both groups. Normality was assessed using Anderson-Darling test. Comparison of categorical variables involved chi-square test or Fisher’s exact test, as appropriate. Survival was estimated by the Kaplan-Meier method.

Each explanatory variable was classified a priori according to 4 domains: baseline characteristics, occurrence of co-existing acute illness, perioperative period characteristics, and the occurrence of postoperative complications. To ensure the validity of this classification, the process was conducted independently by 3 physicians with expertise in perioperative care (J.B., L.Z., B.R.). Full consensus between the experts was required to include a variable within a domain. Correlation between continuous variables was considered significant when the Spearman correlation coefficient was >0.50. The choice between 2 correlated variables was based on their respective clinical relevance. Then, continuous variables were dichotomized by clinically relevant thresholds from the literature or by receiver operating characteristic curve analysis to determine the best threshold for 6-month mortality after HF (maximization of the Youden index).

Assuming a causal link between exposures and mortality, the sum of the exposures averaged attributable fractions (AAF) [20] represents the proportion of death that could be prevented if these exposures could be prevented. To assess the 6-month attributable mortality risk associated with each domain while keeping the final model as simple as possible in the spirit of parsimony, we constructed the final model with 2 simple steps: (1) separately selected the most important variables in each of the 4 domains, and (2) fit the final model with all the variables selected in the first step.

Separately for each domain, all variables were included in a multivariate logistic model with 6-month mortality as the explained variable. Odds ratios (ORs), AAFs [20,21], and their corresponding 95% confidence intervals (95% CIs) were estimated for each variable. For AAFs, CIs were derived with Monte Carlo simulations [20]. The most important variables of each domain-specific multivariate logistic model were selected based on their *p* value (*p* < 0.05). Then, the final multivariate model was constructed with the most important variables of each domain selected in the previous step. No further selection of variables was performed. Again, ORs and AAFs were provided with their 95% CI.

Because the final model depended on how the variables of each domain were selected, 2 sensitivity analyses were performed: (1) selection of the variables with *p* < 0.10 in each domain-specific multivariate logistic model; (2) selection of the 3 variables with the greater AAF in each domain-specific multivariate logistic model.

All models were internally validated with the same approach by using boostrap. Optimism-corrected c-index (i.e., discrimination) and optimism-corrected calibration plots were calculated for each model (Appendix A).

Additional post-hoc statistical analyses were performed to test time trends for mortality, co-existing acute illness, type of fracture, type of surgery, and postoperative complications by Fisher’s test for trend in proportions.

All *p* values are two-tailed, and *p* < 0.05 was considered to denote significant difference. Statistical analyses involved using R v3.6.1.

## 3. Results

Our cohort included 1010 patients (Figure 2); baseline characteristics are shown in Table 1. In total, 32 (3.2%) patients died during acute care, 36 (3.6%) during rehabilitation, and 81 (8.0%) after returning to home and/or an institution. The 6-month mortality rate was 14.8% (95% CI: 12.6 to 17.1). 

We found a significant correlation between the ADL and Rockwood scores (R = −0.68, *p* < 0.001), and IADL and Rockwood scores (R = −0.68, *p* < 0.001). The ADL score was retained as the variable assessing autonomy. We found a correlation between the CIRS score and number of medications (R = 0.57, *p* < 0.001), and the CIRS was retained as the variable assessing comorbidities. The optimal threshold of the ADL score to predict 6-month mortality was a score ≥5.5 (sensitivity 0.57, specificity 0.67). Conversely, the best threshold of the CIRS to predict 6-month mortality was a score ≥11 (sensitivity 0.60, specificity 0.67) and that for age was ≥88 years (sensitivity 0.58, specificity 0.57). In a multivariate model including baselines variables, we determined that the age, gender, CIRS score, ADL score, and type of fracture were significantly associated with 6-month mortality after HF (Appendix A).

Acute co-existing illness (Table 2) was diagnosed in 105 patients (10.4%, 95% CI: 8.6 to 12.4). The occurrence of an acute co-existing illness was not significantly associated with 6-month mortality after HF (Appendix A).

Most patients underwent surgery before 48 h after hospital admission (81.0%, 95% CI: 79.3 to 84.2; median time to surgery: 24 (17–44) h). Perioperative factors are shown in Table 2. On multivariate logistic regression including all perioperative factors, blood transfusion and time to surgery >48 h were associated with 6-month mortality after HF (Appendix A and Appendix A)

Postoperative complications are shown in Table 3; 104 patients (10.3%, 95% CI 8.5 to 12.4) had at least one severe postoperative complication (Dindo-Clavien score ≥ 3) (Appendix A). On multivariate logistic regression, the occurrence of severe postoperative complications was significantly associated with 6-month mortality after HF (OR 5.88, 95% CI 3.78 to 9.12, *p* < 0.001) (Appendix A).

Table 4 and Figure 3 show the AAF of the different domains. The weight of baseline characteristics was greatest (62.4%, 95% CI: 50.0 to 74.7%). Perioperative factors, especially blood transfusion, and severe postoperative complications accounted for 24.2% of the 6-month mortality (95% CI: 9.2 to 39.3). Finally, 13.4% (95% CI: 0 to 26.9) of the 6-month mortality rate was not explained by these variables. Sensitivity analysis gave similar conclusions (Appendix A, Figure 3*).*

Additional post-hoc statistical analyses revealed no significant difference over time in 6-month mortality rate (*p* = 0.80), rate of co-existing acute illnesses (*p* = 0.40), type of fracture (*p* = 0.40), and rate of postoperative complications (*p* = 0.70).

## 4. Discussion

In estimating the respective influence of a priori selected risk factors on six-month mortality after HF, baseline characteristics were the most important contributing factors (62.4%, 95% CI: 50.0 to 74.7%). Postoperative complications (11.9%, 95% CI: 6.9 to 16.9%), perioperative blood transfusion (9.6%, 95% CI: 1.3 to 20.5%) and delayed surgery (2.7%, 95% CI: 1.8 to 7.3%) had lower but still significant weight. Our results, estimating for the first time the respective influence of a priori selected risk factors on six-month mortality after HF, suggest that a maximum of 24.2% of deaths could be avoided if all of these modifiable factors could be prevented.

Baseline characteristics explained less than two-thirds of the six-month mortality. This result helps in understanding why preoperative scores are inaccurate to predict mortality [8] and why mortality after HF is six times that observed after elective total hip replacement, even when patients are matched for baseline characteristics [10]. Although this study was retrospective, its observational nature allowed us to include more patients with preoperative severe baseline characteristics than what we could observe in a randomized controlled trial. Indeed, clinical trials feature an under-representation of older adults, particularly those with multimorbidity and polypharmacy, as compared with actual conditions of medicine use in real-world practice [22]. We were able to obtain a lot of information concerning baseline characteristics of our patients, in line with a geriatric point of view that tries to assess all variables that characterize the geriatric patient versus healthy aging [23]. However, we observed high correlations between these variables, and we finally retained only age and the CIRS score for comorbidities and the ADL score for autonomy.

Delayed surgery was not an important issue (estimates of the six-month attributable mortality: 2.7%), which contrasts with previous studies. A meta-analysis including 191,873 patients [12] found that early surgery (cut-off between 24 and 48 h) was associated with significantly reduced risk of death (OR 0.74; 95% CI: 0.67–0.81). Nevertheless, this previous observational study failed to identify patients with surgery delayed for valid medical reasons, who are thought to have a poorer outcome, thereby suggesting possible confounders. In our study, only 19% of the patients had delayed surgery per this definition, so delay may not be important when HF is considered an urgent surgical procedure and when only a limited number of patients undergo surgery later. However, in most countries, including France, approximately half of the older patients with HF undergo surgery with a delay > 48 h (mainly because of preoperative medical assessment and access to the operating room) [8,24]. A recent large randomized trial failed to observe a significant decrease in mortality when comparing early (median 6 h) vs. delayed (median 24 h) surgery. Our results are consistent with the non-significant estimate of risk reduction (1%, *p* = 0.40) [25].

The type of anesthesia used was not associated with outcome in our study, which agrees with findings from a recent large observational study [26]. Nevertheless, because most of our patients had general anesthesia (96%), our study lacks the power to answer this question. We assessed the perioperative period with only blood transfusion and hemoglobin values, thereby failing to precisely assess perioperative hemodynamic stability. However, there are few data to indicate that intraoperative instability may have a major impact on outcome as compared to other known factors in this population. In a recent meta-analysis of five studies including a total of 403 participants, perioperative hemodynamic optimization did not significantly improve outcomes for older patients with HF [27]. In addition, in a recent multicenter clinical trial of patients predominantly undergoing major abdominal surgery, management targeting individualized blood pressure as compared with standard management reduced the risk of postoperative organ dysfunction but not 30-day mortality [28].

Postoperative complications represent some important factors associated with mortality. This result is consistent with a previous study showing that 57% of in-hospital deaths in older patients with HF may be preventable [29]. This result is also impressive when considering that our study was performed in a dedicated environment where mortality is reduced.^3^ Thus, future research should focus on preventing these complications. Our study suggests that swallowing disorders, postoperative delirium, pressure sores, heart failure, acute myocardial infarction, and atrial fibrillation, among other factors, might require particular attention.

### Limitations

This was an observational study, and causality cannot be demonstrated. Although we considered as much information as possible and used advanced statistical methods to correct for confounding biases, we can only assume that all relevant confounders have been identified. Moreover, the variable selection approach used could be a limitation to the generalization of our findings. However, our sensitivity analyses gave similar results.

More importantly, 13.4% of the deaths observed at six months were not explained by this model, meaning by one of the four domains. Similar results are common in most other clinical predictive models. This finding suggests that our models’ predictive performances were excellent but not perfect. The most relevant interpretation is that some predictors were omitted in the models (i.e., not recorded or unknown predictors), that all events (death) were not preventable, or also that our parsimonious modeling approach wasted a part of the information. An example of omitted predictors could be a defect in immune regulation, as it has been reported in a model of septic acute stress [30]. Indeed, an increase in inflammatory markers has been reported after HF and was associated with post-operative mortality [31,32]. However, while not perfect, the predictive performance of our model supported of modelling approach and allowed us to provide meaningful clinical interpretations without a significant level of imprecision.

Our study was conducted in a highly specialized environment that is associated with reduced six-month mortality as compared with patients admitted to orthopedic departments [9]. Therefore, our results may not be extrapolated to conventional or other orthogeriatric models previously reported.

## 5. Conclusions

Baseline characteristics of the older patients explained less than two-thirds of the six-month mortality after HF surgery. Optimizing the care of older patients with HF, by improving management of perioperative factors (postoperative complications, perioperative blood transfusion, and delayed surgery) could reduce by a maximum of one quarter the six-month mortality rate after HF. These results, providing new information to help design future research at the forefront of care improvement for older patients with hip fracture, indicate that physicians should focus on early detection and treatment of severe postoperative complications. In addition, a 13.4% mortality rate was not explained by our model, indicating that there are still unknown predictive factors.

## Figures and Tables

**Figure 1 jcm-09-02370-f001:**
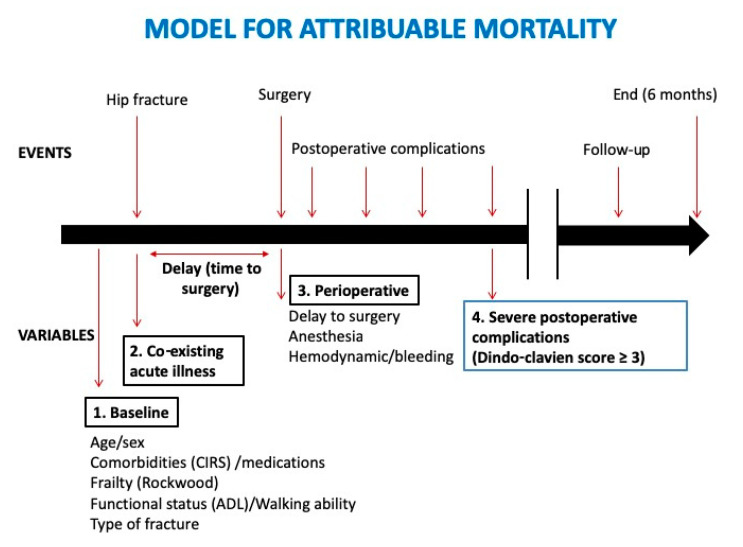
Clinical model for estimating attributable mortality in older patients with hip fracture. Four domains of risk factors (numbered 1 to 4) were investigated. Abbreviations: ADL = Activities of Daily Living; CIRS: Cumulative Illness Rating Scale.

**Figure 2 jcm-09-02370-f002:**
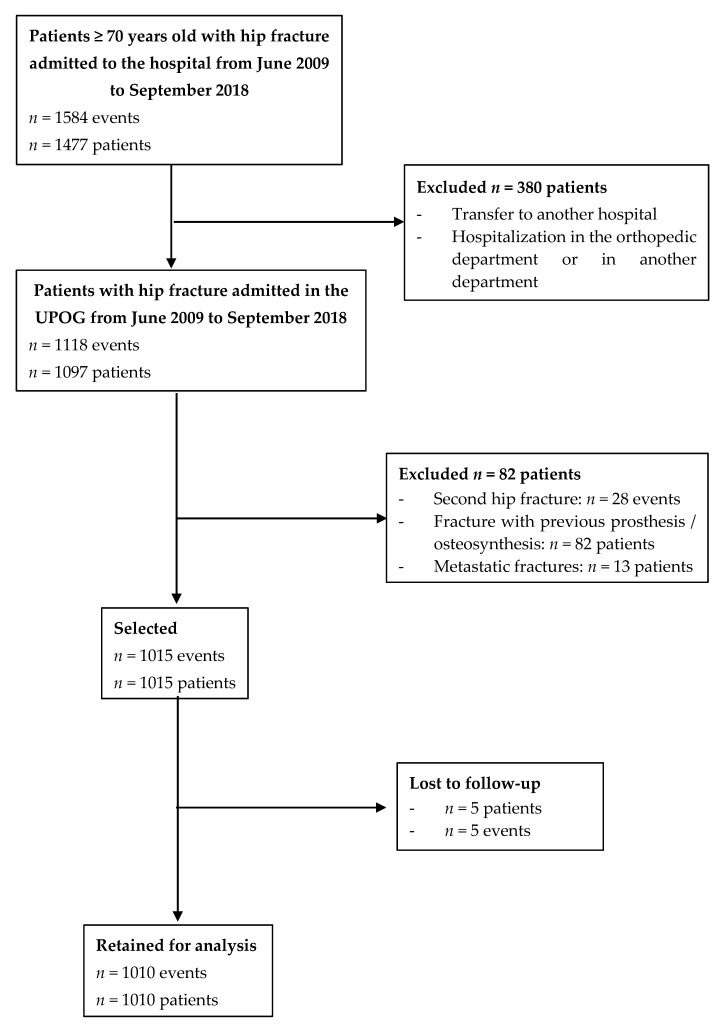
Flow chart.

**Figure 3 jcm-09-02370-f003:**
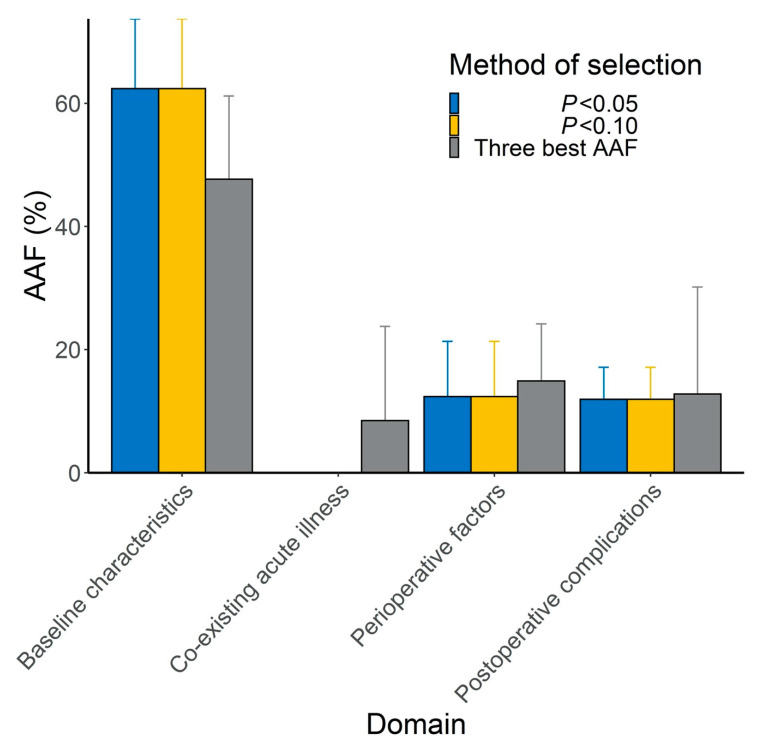
Results of 3 analyses of the estimation of attributable mortality (scale from 0 to 80) associated with baseline characteristics, co-existing acute illness, perioperative factors, and severe postoperative complications. The main analysis was selection of the variables with p < 0.05 for each domain-specific multivariate logistic model and the 2 others were considered sensitivity analyses. The whiskers represent the upper limit of the bilateral 95% confidence interval. Abbreviation: AAF = averaged attributable fraction.

**Table 1 jcm-09-02370-t001:** Baseline characteristics of patients overall and stratified by living status at 6 months after surgery.

	All Patients (*n* = 1010)	Deceased (*n* = 149)	Alive (*n* = 861)	*p* Value
Age (years)	86 ± 6	88 ± 6	86 ± 6	<0.001
Female	768 (76)	99 (66)	669 (78)	0.003
Medical history				
CIRS score	9 (6–12)	12 (9–14)	9 (6–12)	<0.001
CIRS score ≥ 11	378 (37)	90 (60)	288 (33)	<0.001
Dementia	394 (39)	70 (47)	324 (38)	0.03
Stroke	164 (16)	32 (21)	132 (15)	0.06
Hypertension	692 (69)	108 (72)	584 (68)	0.26
Diabetes	139 (14)	26 (17)	114 (13)	0.24
Atrial fibrillation	271 (27)	72 (48)	199 (23)	0.44
Coronary artery disease	183 (18)	39 (26)	144 (17)	0.006
Heart failure	165 (16)	50 (34)	115 (13)	<0.001
Obesity (BMI ≥ 30 kg/m^2)^)	66 (7)	13 (9)	53 (6)	0.24
Chronic renal failure ^a^ Missing values	115 (11) 3	34 (23) 0	81 (9) 3	<0.001
Cancer	222 (22)	49 (33)	173 (20)	<0.001
Number of drugs	5 (3–8)	6 (4–9)	5 (3–7)	<0.001
Anticoagulant	189 (19)	44 (30)	145 (17)	0.002
Antiplatelet	365 (36)	53 (36)	312 (36)	0.87
Autonomy and frailty				
ADL score	5.5 (3.5–6)	4.5 (2.5–5.5)	5.5 (4–6)	<0.001
ADL score > 5.5	543 (54)	49 (33)	494 (57)	<0.001
Rockwood score	5 (4–6)	6 (5–6)	5 (4–6)	<0.001
Living in nursing home	139 (14)	27 (18)	112 (13)	0.09
Walking ability				
Walking without assistance	410 (48)	35 (23)	445 (44)	<0.001
Walking with assistance	544 (54)	110 (74)	434 (50)	<0.001
Not walking	17 (2)	4 (2)	21 (2)	0.54
Fracture				
Intertrochanteric fracture	530 (52)	92 (62)	438 (51)	0.01
Femoral neck fracture	480 (48)	57 (38)	423 (49)	0.01

Data are mean ± SD, median (25–75 interquartile range), or number (percentage). Missing values are detailed only when they exist. Comparison between the two groups by Mann-Whitney U test for quantitative variables and chi-square test for qualitative variables. Abbreviations: ADL = Activities of Daily Living; BMI: body mass index; CIRS: Cumulative Illness Rating Scale. ^a^: see text for definition.

**Table 2 jcm-09-02370-t002:** Co-existing acute illness before surgery and perioperative factors: overall and stratified by living status at 6 months after surgery.

	All Patients (*n* = 1010)	Deceased (*n* = 149)	Alive (*n* = 861)	*p* Value
Acute Co-Existing Illness ^a^				
Another associated trauma	49 (5)	2 (1)	47 (6)	0.003
Any acute co-existing illness	105 (10) ^b^	16 (11)	89 (10)	0.88
Perioperative Factors				
Anesthesia				
General anesthesia	872 (96)	130 (96)	742 (96)	0.80
Missing values	106	14	92	
Surgery				
Time to surgery > 48 h	192 (19)	39 (26)	153 (18)	0.02
Gamma nail	513 (51)	86 (58)	427 (50)	0.07
Dynamic hip screw	65 (6)	8 (5)	57 (7)	0.57
Unipolar prosthesis	406 (40)	54 (36)	352 (41)	0.28
Bipolar prosthesis	26 (3)	1 (0.7)	25 (3)	0.16
Hemoglobin level				
Preoperative hemoglobin (g·dL^−1^)	12.2 ± 1.6	11.5 ± 1.7	12.3 ± 1.5	<0.001
In-hospital ^c^ transfusion	507 (50)	100 (67)	407 (47)	<0.001
In-hospital ^c^ total packed RBC (unit)	1 (0–2)	2 (0–3)	0 (0–2)	<0.001

Data are mean ± SD, median (25–75 interquartile range), or number (percentage). Missing values are detailed only when they exist. Comparison between the two groups by Mann-Whitney U test for quantitative variables and chi-square test or Fisher’s exact test for qualitative variables. Abbreviations: RBC: red blood cell. ^a^: see text for definition. ^b^: trauma lesions (*n* = 49), infections (*n* = 17, mostly pulmonary, *n* = 9, and urinary, *n* = 5, infections), acute cardiac disease (*n* = 12), stroke and seizures (*n* = 7), blood transfusion (*n* = 7), rhabdomyolysis (*n* = 6) and thromboembolic disease (*n* = 3). The sum of conditions may not add to the total because a patient may have several conditions. ^c^: i.e., in the emergency room, surgery, intensive care unit, and perioperative geriatric unit (excluding rehabilitation).

**Table 3 jcm-09-02370-t003:** Postoperative factors: overall and stratified by living status at 6 months after surgery.

.	All Patients (*n* = 1010)	Deceased (*n* = 149)	Alive (*n* = 861)	*p* Value
In-hospital ^a^ postoperative complications				
Dindo-Clavien score	2 (1–2)	2 (2–4)	2 (1–2)	<0.001
Dindo-Clavien score ≥ 3	104 (10)	45 (30)	59 (7)	<0.0001
Postoperative delirium	404 (40)	79 (53)	325 (38)	0.004
Atrial fibrillation	83 (8)	19 (13)	64 (7)	0.03
Acute coronary syndrome	83 (8)	25 (17)	58 (7)	<0.001
Acute heart failure	117 (12)	43 (29)	74 (9)	<0.001
Venous thromboembolic event	44 (4)	6 (4)	38 (4)	0.83
Hemorrhage	93 (9)	21 (14)	72 (8)	0.03
Infection	168 (17)	44 (30)	124 (14)	<0.001
Surgical revision	19 (2)	4 (3)	15 (2)	0.51
Bladder retention	268 (27)	50 (34)	218 (25)	0.04
Stool impaction	448 (44)	76 (51)	372 (43)	0.07
Pressure sore	110 (11)	34 (23)	76 (9)	<0.001
Admission to ICU	47 (5)	20 (13)	27 (3)	<0.001
Walking ability				
Time to first sitting (days) Missing values	1 (1–2) 20	2 (1–3) 7	1 (1–2) 13	0.001
Time to first walking (days) Missing values	2 (1–3) 77	2 (1–4) 22	2 (1–3) 55	0.007
At discharge				
Home ^b^	152 (15)	19 (13)	133 (15)	0.39
Rehabilitation care Missing values	814 (81) 1	93 (63) 1	721 (84) 0	<0.001

Data are mean ± SD, median (25–75 interquartile range), or number (percentage). Missing values are detailed only when they exist. Comparison between the two groups by Mann-Whitney U test for quantitative variables and chi-square test or Fisher’s exact test for qualitative variables. Abbreviations: ICU: intensive care unit. ^a^: i.e., in the emergency room, surgery, ICU, and perioperative geriatric unit (excluding rehabilitation). ^b^: home includes institution if the patient was previously in an institution.

**Table 4 jcm-09-02370-t004:** Estimates of the 6-month attributable mortality for each domain.

Variables	OR (95% CI)	*p* Value	AAF (%) (95% CI)
Baseline characteristics → AAF = 62.4			
Age, ref < 88 years • Age ≥ 88 years	1.72 (1.16–2.56)	0.007	10.8 (2.9–18.7)
Gender, ref = Female • Gender = Male	1.93 (1.25–2.97)	0.003	7.3 (1.4–13.1)
CIRS score, ref < 11 • CIRS score ≥ 11	2.27 (1.55–3.35)	<0.001	15.6 (6.1–25.0)
ADL score, ref ≥ 5.5 • ADL score < 5.5	2.52 (1.68–3.76)	<0.001	20.0 (9.8–30.2)
Femoral neck fracture, ref = No • Femoral neck fracture = Yes	0.67 (0.45–1.01)	0.054	8.7 (0.1–17.2)
Co-existing acute illness → AAF = 0			
Perioperative factors → AAF = 12.3			
Time to surgery, ref ≤ 48 h • Time to surgery > 48 h	1.36 (0.87–2.13)	0.18	2.7 (1.8–7.3)
Transfusion, ref = No • Transfusion = yes	1.53 (1.02–2.28)	0.04	9.6 (1.3–20.5)
Postoperative factors →AAF = 11.9			
Dindo-Clavien score, ref < 3 • Dindo-Clavien score ≥ 3	4.91 (3.06–7.90)	< 0.001	11.9 (6.9–16.9)
		Total	86.6 (73.1–100)

*n* = 1010, C-Index = 0.78 95% CI (0.74–0.82); Hosmer–Lemeshow test: X2 = 10.737, ddl = 8, *p* = 0.22. Co-existing acute illness factors were not included in the final model because they were not significant on univariate analysis. Abbreviations: AAF = averaged attributable fraction, ADL: Activities of Daily Living, CIRS: Cumulative Illness Rating Scale; OR = odds ratio, CI: confidence interval; ref = reference value.

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
