# Peer review of "Attributable Mortality of Hip Fracture in Older Patients: A Retrospective Observational Study"

_jcm, 2020, doi:10.3390/jcm9082370_

Round 1

Reviewer 1 Report

The  manuscript “Attributable mortality of hip fracture in older patients: a cohort study in a dedicated orthogeriatric care pathway” by Lorène Zerah et al. aimed to estimate the respective influence of four domains including the baseline characteristics, co-existing acute illnesses, perioperative factors and postoperative complications on 6-month mortality after hip fracture. Baseline characteristics of the patients explained less than two thirds of the 6-month mortality after hip fracture surgery. Optimizing the care of older patients by improving management of perioperative factors could reduce by a maximum of a quarter the 6-month mortality rate after hip fracture.  

COMMENTS

  1. The Authors should describe the 4 domains and reported Figure 1 in the “Materials and Methods” section.
  2. The Table 1 is too long; it would be better to separate the domains into different tables.
  3. In Table 1 also BMI value or nutritional status should be reported.
  4. The information on previous fracture history (vertebral, wrist, humerus) were reported.
  5. The “discussion” section seems to be short, in particular the results of this study should be better discussed and compared with previous literature.
  6. English grammar and syntax should be improved.

Author Response

ITEM-BY –ITEM RESPONSE TO THE REVIEWER 1

Attributable Mortality of Hip Fracture in Older Patients: a cohort study in a dedicated orthogeriatric care pathway

Manuscript ID: jcm-853804

We appreciate the opportunity to address Reviewers’ comments and revise our manuscript accordingly. Below, please find item-by-item responses to the Reviewer 1’ comments, which are included verbatim.  All page and paragraph numbers refer to locations in the revised manuscript.

COMMENT 1: The Authors should describe the 4 domains and reported Figure 1 in the “Materials and Methods” section.

RESPONSE:

  • As proposed by the reviewer, we have shortened the introduction to describe the 4 domains in the “Materials and Methods” section (with Figure 1).
  • In the introduction section, the description is now All these factors could be classified into 4 domains: baseline characteristics, co-existing acute illnesses, perioperative factors and postoperative complications.” (Page 3, Lines 63-65).
  • In the “Materials and Methods” section, we have added a subsection “2.2 Description of the 4 domains”, with the initial description and the Figure 1 (Page 4, Lines 91-97).

COMMENT 2: The Table 1 is too long; it would be better to separate the domains into different tables.

RESPONSE:

  • As proposed by the reviewer, we have separated the initial Table 1 into 3 tables:
    • Table 1 : Patient baseline characteristics: overall and stratified by living status at 6 months after surgery (N = 1010) (Page 9)
    • Table 2: Co-existing acute illness before surgery and perioperative factors: overall and stratified by living status at 6 months after surgery (N = 1010) (Page 11)
    • Table 3: Postoperative factors: overall and stratified by living status at 6 months after surgery (N = 1010) (Page 12)

COMMENT 3: In Table 1 also BMI value or nutritional status should be reported.

RESPONSE:

  • As proposed by the reviewer, we have added the Obesity status (Body Mass Index (BMI) ≥ 30 kg/m2) in the Table 1 (Page 9).
  • We also have the data for the level of albumin, a marker of nutritional status. We chose not to include it in Table 1 because this marker is influenced by inflammation, which is often high at the time of the fracture. If however you think it is useful to add it, here are the data.

All patients

(N=1010)

Deceased

(N=149)

Alive

(N=861)

P value

Albumin

Missing values

29 [26-32]

36

29 [26-31]

8

29 [26-32]

28

<.001

Data are median [25–75 interquartile range]

Comparison between groups by Mann-Whitney U test for quantitative variables

COMMENT 4: The information on previous fracture history (vertebral, wrist, humerus) were reported.

RESPONSE:

We agree that this information may have improved description of bone vulnerability and osteoporosis of patients. However, hip fracture is a sufficient criteria to consider a severe form of osteoporosis, and unfortunately,  we do not have this information.

COMMENT 5: The “discussion” section seems to be short, in particular the results of this study should be better discussed and compared with previous literature.

RESPONSE:

We thank the reviewer for this comment, and we have tried to better discuss and compare our results with previous literature. We have added new paragraphs and references for the baseline characteristics (page 16, lines 285–293, references 22,23), delayed surgery (page 16, lines 295–299, lines 302-304, references 24), type of anesthesia (page 16, lines 308-310, references 26), perioperative hemodynamic optimization (page 17, lines 3124–319, reference 28), and postoperative complications (page 17, lines 320-326, reference 29).

COMMENT 6: English grammar and syntax should be improved.

RESPONSE:

We had our manuscript checked by a professional English editor (Mrs Laura Smales from BioMedEditing, Toronto, Canada) (Page 20, Lines 395)

Reviewer 2 Report

This is a well written analysis of potential risk factors for mortality after hip fracture in a specialized unit for orthogeriatric care. A large amount of data was analyzed. I only have the suggestion to improve the abstract. The following paragraph is not clear immediately and should be rewritten:

The 6-month mortality rate was 14.8 %. Estimates of the 6-month attributable mortality were as follows: baseline characteristics (including age, gender, comorbidities, autonomy, type of fracture): 62.4%; co-existing acute illnesses: 0%; perioperative factors (including blood transfusion and delayed surgery): 12.3%; severe postoperative complications: 11.9%.

co-existing acute illnesses: 0% shoud be shortly explained why 0%

Summarized, thank you for this important data.

Author Response

ITEM-BY –ITEM RESPONSE TO THE REVIEWER 2

Attributable Mortality of Hip Fracture in Older Patients: a cohort study in a dedicated orthogeriatric care pathway

Manuscript ID: jcm-853804

We appreciate the opportunity to address Reviewers’ comments and revise our manuscript accordingly. Below, please find item-by-item responses to the Reviewer 2’ comments, which are included verbatim.  All page and paragraph numbers refer to locations in the revised manuscript.

COMMENT: This is a well written analysis of potential risk factors for mortality after hip fracture in a specialized unit for orthogeriatric care. A large amount of data was analyzed. I only have the suggestion to improve the abstract. The following paragraph is not clear immediately and should be rewritten:

The 6-month mortality rate was 14.8 %. Estimates of the 6-month attributable mortality were as follows: baseline characteristics (including age, gender, comorbidities, autonomy, type of fracture): 62.4%; co-existing acute illnesses: 0%; perioperative factors (including blood transfusion and delayed surgery): 12.3%; severe postoperative complications: 11.9%.

co-existing acute illnesses: 0% shoud be shortly explained why 0%

Summarized, thank you for this important data.

RESPONSE:

We thank the reviewer for her/his comments. As proposed, we added a brief explanation of what co-existing acute illnesses were and why the 6-month mortality estimate attributable to co-existing acute illnesses was 0%: co-existing acute illnesses (which include acute events present before surgery that could result from the fracture or cause it): 0% (not significantly associated with 6-month mortality)” (Page 1-2, Lines 37-38).

Reviewer 3 Report

This paper presents a retrospective observational study using a monocentric cohort of older patients to evaluate the respective influence of these four domains on six-month mortality in patients suffering an HF. While the paper presents the right amount of work, hypothesis, discussion, and take-home message are not clear.

Major comments:

Introduction:

The last part of the introduction should present a clear foreground question. Furthermore, a hypothesis should be presented in terms of major co-factors and minor co-factors.

Method:

The dedicated research database, which was prospectively supplemented by three senior geriatricians (JB, JCB, LZ), however, in case of disagreement between the two primary adjudicators regarding Co-existing illness, reconciliation was reached with a third senior expert (BR). Please clarify why BR was only involved in this parameter, and only one person evaluated the other parameters. It needs significant clarification to disclose a bias in the data collection.

Stats:

please provide which normativity test was used and in which case the data were normally distributed or not. A parametric test provides a quantitative difference when a non-parametric test provides a qualitative difference. Discuss the results accordingly. Same for correlation, please use Pearson in case of normal distribution and Spearman in case of non-normal distribution.

Provide the results of the distribution in a supplemental table.

Why choosing variable with p<0.1, please justify.

Who was the person performing the stats ? was this person considered a blinded investigator? If so, please explain how the data were blinded.

Discussion:

Because co-morbidity severity, frailty, and functional status were calculated by one of the three senior geriatricians during the hospital stay, it is interesting to discuss the results by providing the inter-rater and intra-rater variability of each scale used.

The discussion is relatively short when looking at all the data and results available. The take-home message is not clear as well. What are the major factors, what are the minor factors? What is the reliability of their evaluations?

It should be interesting to mention what are the possible postoperative complications that occur in the rehabilitation unit for guiding the next study.

minor comments:

the title could be shorter and include the retrospective aspect

"Attributable mortality of hip fracture in older patients: a retrospective observational study."

Please provide quantitative information, such as the number of people affected in Europe, or the percentage of the population in a specific class of age.

line 55: "6 times higher than."

English editing (not only grammar) could be interesting to avoid this succession of mini-paragraph and connect ideas better.

Author Response

ITEM-BY –ITEM RESPONSE TO THE REVIEWER3

Attributable Mortality of Hip Fracture in Older Patients: a cohort study in a dedicated orthogeriatric care pathway

Manuscript ID: jcm-853804

We appreciate the opportunity to address Reviewers’ comments and revise our manuscript accordingly. Below, please find item-by-item responses to the Reviewer 2’ comments, which are included verbatim.  All page and paragraph numbers refer to locations in the revised manuscript.

COMMENT 1: INTRODUCTION

The last part of the introduction should present a clear foreground question. Furthermore, a hypothesis should be presented in terms of major co-factors and minor co-factors.

RESPONSE:

  • As proposed by the reviewer, we have modified the end of the introduction to present a clear foreground question and a hypothesis: All these factors could be classified into 4 domains: baseline characteristics, co-existing acute illnesses, perioperative factors and postoperative complications. What are their respective influence on 6-month mortality? We hypothesized that understanding their respective influence on 6-month mortality and identifying the modifiable factors with the highest impact is an essential step that may indicate the directions for care improvement.” (Page 3 , Lines 63-68).
  • As our primary objective was to evaluate the respective influence of these 4 domains on 6-month mortality in patients after HF, we could not present them in terms of major co-factors or minor co-factors, because we did not know, before the study, which would be minor or major. Moreover, this presentation was not a priori decided in the statistical analysis and should be considered as a post hoc presentation.

COMMENT 2: METHOD

The dedicated research database, which was prospectively supplemented by three senior geriatricians (JB, JCB, LZ), however, in case of disagreement between the two primary adjudicators regarding Co-existing illness, reconciliation was reached with a third senior expert (BR). Please clarify why BR was only involved in this parameter, and only one person evaluated the other parameters. It needs significant clarification to disclose a bias in the data collection.

RESPONSE:

  • We have created a dedicated research database that was prospectively supplemented by 3 senior geriatricians (JB, JCB, LZ) and that integrates all the data from the orthogeriatric care pathway for each patient. Therefore, co-morbidity severity, frailty and functional status were calculated prospectively by one of the 3 senior geriatricians during the hospital stay (Page 5, Lines 104 - 106).
  • Co-existing acute illnesses and the severity of postoperative complications were the only 2 variables to have been classified retrospectively. They were both adjudicated by 2 senior geriatricians (JB and LZ), independently reviewing medical charts. In case of disagreement, reconciliation was reached with a third independent senior expert (BR). We added this clarification in the revised manuscript (Page 6, Lines 124 - 128).

COMMENTS3: STATISTICS

Please provide which normativity test was used and in which case the data were normally distributed or not. A parametric test provides a quantitative difference when a non-parametric test provides a qualitative difference. Discuss the results accordingly.

RESPONSE:

  • For better clarity, we replaced this paragraph in the statistics section : Comparison between survivors and deceased patients involved unpaired Student t test, Mann-Whitney test, chi-square test or Fisher’s exact test, as appropriate” by : “Comparison of quantitative variables between survivors and deceased patients involved unpaired Student t tests, or Mann-Whitney tests in case of rejection of the normality assumption in one or both groups. Normality was assessed using Anderson-Darling test. Comparison of categorical variables involved chi-square test or Fisher's exact test, as appropriate." (Page 6, Lines 142-147).
  • In fact, for all comparisons of quantitative variables between survivors and deceased patients, a non-parametric tests was finally used because of the rejection of the normality assumption. This is now specified at the end of each table (page 9, 11, 12).

Same for correlation, please use Pearson in case of normal distribution and Spearman in case of non-normal distribution.

RESPONSE:

All correlations were estimated again using Spearman correlation coefficients, because this approach is appropriate for both normal and non-normal distribution (Page 6, Lines 152-155). Results were almost unchanged (modification of the second decimal) (Page 10, Lines 202-205).

Why choosing variable with p<0.1, please justify.

RESPONSE:

The primary analysis was performed with variables with p<0.05. Because the choice of the variable selection algorithm may have an impact on the final multivariate model, sensitivity analyses were performed to explore the robustness of our results : Because the final model depended on how the variables of each domain were selected, 2 sensitivity analyses were performed: 1) selection of the variables with P<.10 in each domain-specific multivariate logistic model; 2) selection of the 3 variables with the greater AAF in each domain-specific multivariate logistic model.” (Page 7, Lines 173-176).

Who was the person performing the stats ? was this person considered a blinded investigator? If so, please explain how the data were blinded.

RESPONSE:

Dr D Hajage, public health physician and statistician, performed the statistical analyses. He was an independent statistician which was not involved in data collection (Page 19, Lines 380-382). The statistical plan of the study was established before the statistical analysis (Page 6, Lines 139-141)

COMMENTS4 : DISCUSSION

Because co-morbidity severity, frailty, and functional status were calculated by one of the three senior geriatricians during the hospital stay, it is interesting to discuss the results by providing the inter-rater and intra-rater variability of each scale used.

  • We don’t have the data to provide inter-rater or intra-rater variability of each scale used. Data were collected prospectively by one of the 3 physicians, without double collection.
  • We were only able to provide the percentage of agreement (kappa score) between experts (LZ and JB) for acute co-existing illnesses and the Dindo-Clavien score. Indeed, co-existing acute illnesses and the severity of postoperative complications were the only 2 variables to have been classified retrospectively. They were both adjudicated by 2 senior geriatricians (JB and LZ), independently reviewing the medical charts (Kappa score 0.90 for acute co-existing illnesses and 0.97 for Dindo-Clavien score). In case of disagreement, reconciliation was reached with a third independent senior expert (BR) (Page 6, Lines 124-128).

The discussion is relatively short when looking at all the data and results available

RESPONSE

We thank the reviewer for this comment, and we have tried to better discuss and compare our results with previous literature. We have added new paragraphs and references for the baseline characteristics (page 16, lines 285–293, references 22,23), delayed surgery (page 16, lines 295–299, lines 302-304, references 24), type of anesthesia (page 16, lines 308-310, references 26), perioperative hemodynamic optimization (page 17, lines 3124–319, reference 28), and postoperative complications (page 17, lines 320-326, reference 29).

The take-home message is not clear as well. What are the major factors, what are the minor factors? What is the reliability of their evaluations?

RESPONSE

  • We have added in the discussion a first paragraph to synthesize our results explaining what are the major and minor contributing factors (Page 16, lines 275 – 281).
  • The reliability of our evaluations are discussed in the limitations : “This finding suggests that our models’ predictive performances were excellent but not perfect. The most relevant interpretation is that some predictors were omitted in the models (i.e., not recorded or unknown predictors), that all outcomes (mortality) were not preventable, or also that the modeling approach (i.e., stratification, parsimonious approach) wasted a part of the information.” (Page 17, Lines 335-339)

It should be interesting to mention what are the possible postoperative complications that occur in the rehabilitation unit for guiding the next study.

RESPONSE

  • Indeed, it would have been very interesting. Unfortunately, as stated in Material and Methods sections, all postoperative complications during the acute care period (not in rehabilitation) were prospectively recorded (Page 6, Lines 122-123). Therefore, we do not have this information.

COMMENTS5: minor comments

The title could be shorter and include the retrospective aspect. "Attributable mortality of hip fracture in older patients: a retrospective observational study."

RESPONSE

As proposed by the reviewer, we changed the title (Page 1, Lines 3-4).

Please provide quantitative information, such as the number of people affected in Europe, or the percentage of the population in a specific class of age.

RESPONSE:

As proposed by the reviewer, we have added in the introduction this sentence: More than 1.6 million of people undergo hip fracture in the world every year” (Page 3, Line 52-53, reference 3)

line 55: "6 times higher than."

English editing (not only grammar) could be interesting to avoid this succession of mini-paragraph and connect ideas better.

RESPONSE:

We had our manuscript checked by a professional English editor (Mrs Laura Smales from BioMedEditing, Toronto, Canada) (Page 20, Lines 395)

Round 2

Reviewer 3 Report

The authors provided a better version of their manuscript in terms of method and discussion. However, the introduction has to be improved by stating a clear hypothesis. Here, the authors collected lots of data without supporting or testing a hypothesis. This research is presented like a fishing expedition. This is a large amount of work that needs a better presentation. This will facilitate the process of discussing the results as well.

Introduction:
Indeed, the authors should use previous findings of mortality prediction, as I already suggested. For instance, they cite their previous study: "Outcomes after hip fracture surgery compared with elective 426 total hip replacement. JAMA 2015, 314, 1159–1166, doi:10.1001/jama.2015.10842." in the introduction, they should use their findings to investigate the relevant factors as explained below.
For instance, what do they mean by patients are "matched" Please clarify (line 59 ). To my understanding, the goal of matching is to adjust factors by making like-to-like comparisons. What is the link to this study?
Authors stated that when patients are matched for preoperative medical histories, the in-hospital mortality after HF remains six times "higher" than observed after an elective total hip replacement (line 60
remains six times what? I guess "higher" is missing.)
Thus, it means that the baseline characteristics and co-existing acute illnesses have already been studied without success. I guess the authors have to focus on perioperative factors and postoperative complications. If this is not correct, please clarify that point.
To summarize, the link between Line 59 to 61 is not clear; it seems to be pivotal for introducing their research. Authors should provide more references to previous research in this area -- use the ones form your discussion -- and existing theory and use them in relation to their previous findings. Thus, they will evaluate primary and secondary factors. Based on their introduction, the hypothesis must be quantitative and specific (testable with existing data). In this case, it must predict a relationship of a specific size.

I recommend the authors to present their work by testing the following hypothesis:
"perioperative factors and postoperative complications are major predictors of the 6-month mortality rate in the population of older patients with hip fracture".

Method:
In this study, it is difficult to understand who is blinded, who is not blinded, if the data were blinded to the statisticians, etc... Please clarify this aspect.
For instance, the steps to ensure 'quality control' are not described? E.g., Double coding, research team discussion of the identified item, respondent validation.

It seems that most of the steps were not blinded, so all the researcher(s) (including statistician, research assistants.., all the co-authors, basically) should better state their position in relation to the research question. For
example, – their background and existing knowledge or personal experience of the topic to be researched. This should be clearly stated in the methods part.

Discussion:
Based on their results, it seems that the authors will invalidate their hypothesis and previous research by stating that baseline characteristics are major predictors of the 6-month mortality rate in the population of older patients with hip fracture.

Line 334 This is a very interesting aspect. Indeed, based on the results of baseline characteristics, you present that 13.4% were not predicted by the model. Please elaborate on the 'negative cases,' i.e., narratives that do not fit the identified themes/ theoretical framework. For example, you mentioned that you retained only age and the CIRS score for comorbidities and the ADL score for autonomy. Please identify some participants that did not fit your main findings (meaning do you have outliers ? how many people did not fit the prediction ?) Does this 13.4 % used these factors solely? This is pivotal for establishing new policies.

Conclusion
Based on rewriting the hypothesis, please change the conclusion accordingly.

Author Response

ITEM-BY –ITEM RESPONSE TO THE REVIEWER3

Attributable mortality of hip fracture in older patients: a retrospective cohort study Manuscript ID: jcm-853804

The authors provided a better version of their manuscript in terms of method and discussion. However, the introduction has to be improved by stating a clear hypothesis. Here, the authors collected lots of data without supporting or testing a hypothesis. This research is presented like a fishing expedition. This is a large amount of work that needs a better presentation. This will facilitate the process of discussing the results as well.

COMMENT 1 INTRODUCTION:

Indeed, the authors should use previous findings of mortality prediction, as I already suggested. For instance, they cite their previous study: "Outcomes after hip fracture surgery compared with elective 426 total hip replacement. JAMA 2015, 314, 1159–1166, doi:10.1001/jama.2015.10842." in the introduction, they should use their findings to investigate the relevant factors as explained below. 
For instance, what do they mean by patients are "matched" Please clarify (line 59 ). To my understanding, the goal of matching is to adjust factors by making like-to-like comparisons. What is the link to this study?

Authors stated that when patients are matched for preoperative medical histories, the in-hospital mortality after HF remains six times "higher" than observed after an elective total hip replacement (line 60 remains six times what? I guess "higher" is missing.)

Thus, it means that the baseline characteristics and co-existing acute illnesses have already been studied without success. I guess the authors have to focus on perioperative factors and postoperative complications. If this is not correct, please clarify that point. 

To summarize, the link between Line 59 to 61 is not clear; it seems to be pivotal for introducing their research. Authors should provide more references to previous research in this area -- use the ones form your discussion -- and existing theory and use them in relation to their previous findings. Thus, they will evaluate primary and secondary factors. Based on their introduction, the hypothesis must be quantitative and specific (testable with existing data). In this case, it must predict a relationship of a specific size.

I recommend the authors to present their work by testing the following hypothesis:
"perioperative factors and postoperative complications are major predictors of the 6-month mortality rate in the population of older patients with hip fracture".

RESPONSE:

We thank the reviewer for his/her helpful comment. As the reviewer pointed out, we had previously shown that comorbidities, alone, did not fully explain postoperative mortality. However, we did not know what their attributable risk was.

From these previous studies, we hypothesized that other, potentially modifiable factors, affect mortality, including perioperative factors, postoperative factors and co-existing acute illnesses. The demonstration that a dedicated clinical action plan can significantly improve the 6-month mortality of elderly patients with hip fracture (HF), compared to patients admitted to the orthopedic surgery department, is in favor of this hypothesis. But, the relation between these factors and postoperative mortality is complex, and a fuller understanding of the contribution of each factor is needed to develop a better predictive model for hip fracture outcomes in older people. Therefore, we searched on PubMed, on March 1st 2018 and on December 1st 2019, if a study had attempted to quantify the attributable mortality of HF in the elderly, without success [(((attributable mortality[Title/Abstract]) OR (averaged attributable fractions [Title/Abstract])) AND hip fracture[Title/Abstract])].  Our objective was to evaluate the respective influence of these 4 domains on 6-month mortality in patients after HF, and not only the perioperative and postoperative factors. To our knowledge, this is the first study trying to quantify attributable mortality of hip fracture in older patients. This study was exploratory, and this is the reason why our hypothesis was not quantitative.

  • As proposed, we have developed the introduction to further explain our objective (Page 3, Lines 56 - 80) . We hope that the changes made will satisfy you.
  • Clarification was made on “patients are matched”, meaning that “patients are matched for age, sex and preoperative comorbidities” (Page 3, Lines 59-60)
  • Higher was actually missing. Thanks to the reviewer, we have reintroduced it in the revised version (Page 3, 60).
  • Finally, we think that it would not be scientifically appropriate to modify our hypothesis and objective a posteriori. Some scientific authorities have even considered that it could be considered as a scientific misconduct. In our study, we pay attention to decide a priori our objective and the statistical plan associated to this objective.

COMMENT 2 METHOD:

In this study, it is difficult to understand who is blinded, who is not blinded, if the data were blinded to the statisticians, etc... Please clarify this aspect. 
For instance, the steps to ensure 'quality control' are not described? E.g., Double coding, research team discussion of the identified item, respondent validation. It seems that most of the steps were not blinded, so all the researcher(s) (including statistician, research assistants.., all the co-authors, basically) should better state their position in relation to the research question. For
example, – their background and existing knowledge or personal experience of the topic to be researched. This should be clearly stated in the methods part.

RESPONSE:

The statistician was not blinded to the study hypothesis during the analysis. However, Dr D Hajage, public health physician and statistician, who performed the statistical analyses, was an independent statistician which was not involved in data collection or the initial definition of study objectives. The statistical plan of the study was established by him, after discussion with the authors, before transmission of the data and the beginning of the analyses (Page 6, lines 149 – 153; Page 20 lines 400 – 402). To our knowledge, this is the most rigorous way to perform such analyses and we are convinced having followed the more stringent recommendations available to date. 

This retrospective observational cohort study was conducted in the perioperative geriatric unit (UPOG) of an academic hospital. Since the opening of UPOG in 2009, we have created a dedicated research database that was prospectively supplemented by 3 senior geriatricians (JB, JCB, LZ), experts in orthogeriatrics, and that integrates all the data from the orthogeriatric care pathway for each patient (Page 5, Lines 114 – 116). As the database was prospectively supplemented, all the authors were “blinded” to the research question at the time of data collection (Page 6, Lines 152 – 153).

As explained in the previous review, data were collected prospectively by one of the 3 physicians, without double collection. Co-existing acute illnesses and the severity of postoperative complications were the only 2 variables to have been classified retrospectively for this specific study. They were both adjudicated by 2 senior geriatricians (JB and LZ), independently reviewing the medical charts (Kappa score 0.90 for acute co-existing illnesses and 0.97 for Dindo-Clavien score). In case of disagreement, reconciliation was reached with a third independent senior expert (BR) (Page 6, Lines 134-138). This data collection was performed before any statistical analysis.

COMMENT 3 DISCUSSION:

Based on their results, it seems that the authors will invalidate their hypothesis and previous research by stating that baseline characteristics are major predictors of the 6-month mortality rate in the population of older patients with hip fracture.

RESPONSE:

As explained in Comment 1, we think that it would not be scientifically appropriate to modify our hypothesis and objective a posteriori. Therefore, we did not modify the discussion and the conclusion.

Our objective was to evaluate the respective influence of these 4 domains on 6-month mortality in patients after HF. The results to this objective were : “In estimating the respective influence of a priori selected risk factors on 6-month mortality after HF, baseline characteristics were the most important contributing factors (62.4%, 95% CI: 50.0 to 74.7%). Postoperative complications (11.9%, 95% CI: 6.9 to 16.9%), perioperative blood transfusion (9.6%, 95% CI: 1.3 to 20.5%) and delayed surgery (2.7%, 95% CI: 1.8 to 7.3%) had lower but still significant weight. Our results, estimating for the first time the respective influence of a priori selected risk factors on 6-month mortality after HF, suggest that a maximum of 24.2% of deaths could be avoided if all of these modifiable factors could be prevented”. (Page 17, Lines 289 – 295).

COMMENT 4 DISCUSSION:

Line 334 This is a very interesting aspect. Indeed, based on the results of baseline characteristics, you present that 13.4% were not predicted by the model. Please elaborate on the 'negative cases,' i.e., narratives that do not fit the identified themes/ theoretical framework. For example, you mentioned that you retained only age and the CIRS score for comorbidities and the ADL score for autonomy. Please identify some participants that did not fit your main findings (meaning do you have outliers ? how many people did not fit the prediction ?) Does this 13.4 % used these factors solely? This is pivotal for establishing new policies.

RESPONSE:

  • We thank the reviewer for his/her comment. Indeed, 13.4% of the deaths observed at 6 months were not explained by this model, meaning by one of the four domains (Page 18, Lines 348-349).
  • The most relevant interpretation is that some predictors were omitted in the models (e., not recorded or unknown predictors), that all events (deaths) were not preventable, or also that our parsimonious modeling approach wasted a part of the information (Page 19 Lines 350 – 353).
  • As proposed by the reviewer, we added the possible omitted predictors : “An example of omitted predictors could be a defect in immune regulation, as it has been reported in a model of septic acute stress.[30] Indeed, an increase in inflammatory markers has been reported after HF and was associated with post-operative mortality.[31,32]” (Page 18, Lines 353 – 356).
  • In addition, we added this sentence in the conclusion: “In addition, a 13.4% mortality rate was not explained by our model, indicating that there are still unknown predictive factors.” (Page 19, Line 371-372).
  • Nevertheless, we would like to highlight to the reviewer that a model which explain 100% of death events would be a highly unlikely result (not to say doubtful).
  • Lastly, it should be considered that using this methodology, it would have been very surprising to not find a small proportion of “unexplained” deaths.

COMMENT 5 CONCLUSION:

Based on rewriting the hypothesis, please change the conclusion accordingly.

RESPONSE:

We are sorry, but as explained in Comments 1 and 3, we have not rewritten the hypothesis. However, we have modified the conclusion based on Comment 4 (Page 19, Lines 371-372). We hope that the information provided enables you to understand this choice.